# More journal articles and fewer books: Publication practices in the social sciences in the 2010's

**William E. Savage, Anthony J. Olejniczak** *

Academic Analytics Research Center (AARC), Columbus, Ohio, United States of America

* aolejniczak@aarcresearch.com

## Abstract

The number of scholarly journal articles published each year is growing, but little is known about the relationship between journal article growth and other forms of scholarly dissemination (e.g., books and monographs). Journal articles are the de facto currency of evaluation and prestige in STEM fields, but social scientists routinely publish books as well as articles, representing a unique opportunity to study increased article publications in disciplines with other dissemination options. We studied the publishing activity of social science faculty members in 12 disciplines at 290 Ph.D. granting institutions in the United States between 2011 and 2019, asking: 1) have publication practices changed such that more or fewer books and articles are written now than in the recent past?; 2) has the percentage of scholars actively participating in a particular publishing type changed over time?; and 3) do different age cohorts evince different publication strategies? In all disciplines, journal articles per person increased between 3% and 64% between 2011 and 2019, while books per person decreased by at least 31% and as much as 54%. All age cohorts show increased article authorship over the study period, and early career scholars author more articles per person than the other cohorts in eight disciplines. The article-dominated literatures of the social sciences are becoming increasingly similar to those of STEM disciplines.

## Introduction

The number of scientific and scholarly journal articles published each year has been increasing for some time. Kyvik [1] estimated there was a 30% increase in scientific and scholarly publishing between 1980 and 2000. In a later study, Kyvik and Aksnes [2] noted that Web of Science records increased from 500,000 indexed articles in 1981 to 1.5 million indexed articles in 2013. In 2018 Johnson et al. [3] estimated that the number of scholarly journals grew 5–6% per year over the past decade, and that there were 33,100 peer-reviewed English language journals distributing approximately 3 million articles each year. Much less attention has been given to the scholarly production of books, and the potential relationship between increased journal article publishing and book publishing practices. The social sciences in particular are comprised of several disciplines in which scholars regularly publish both journal articles and books,

**Data Availability Statement:** All anonymized data, scripts, and statistical results files are available from the OSF database (https://osf.io/2x4uf/, DOI 10.17605/OSF.IO/2X4UF).

**Funding:** WES and AJO received data and computing resources from Academic Analytics, LLC (http://www.academicanalytics.com). The funders had no role in study design, data collection and analysis, decision to publish, or preparation of the manuscript.

**Competing interests:** AJO and WES are paid employees of Academic Analytics, LLC. None of the authors has an equity interest in Academic Analytics, LLC. The results presented reflect the authors' opinions, and do not necessarily reflect the opinions or positions of Academic Analytics, LLC. Academic Analytics, LLC management had no oversight or involvement in the project and were not involved in preparation or review of the manuscript. All work was done as part of the respective authors' research, with no additional or external funding. This does not alter our adherence to PLOS ONE policies on sharing data and materials.

representing a unique opportunity to explore the growth of journal article publications in a sample of disciplines where journal articles are not the sole preferred mode of knowledge dissemination.

In this study, we examine how disciplinary publishing practices in the social sciences have changed over a recent nine-year period at Ph.D. granting universities in the United States. We begin by reviewing two of several proposed factors that may underlie the increase in journal article publication in the social sciences. We then quantify changes in total and per capita journal article and book publication output of each discipline, as well as the rate of participation in both modes of dissemination (i.e., the percentage of scholars who have participated in journal article and book authorship over time). Both the rate of publication and the rate of participation are compared across academic age groups (early career, mid-career, and senior scholars). Specifically, we address the following questions using a large database of scholarly activity spanning several social sciences disciplines and including tens of thousands of individual scholars:

1. Has the publication strategy of social science disciplines changed such that more or fewer books and journal articles are being written now than in the recent past (in total and in the context of faculty population changes in these disciplines over time)?

2. Has the percentage of scholars actively participating in a particular publishing type changed over time (e.g., are fewer scholars authoring books, or are fewer books being published per scholar, or both)?

3. Do different faculty age groups show different publication strategies?

## Changes in the social science research environment and performance-based measures

Within the context of overall growth in scientific and scholarly article publication rates, changes in social science journal article publishing have been studied by several researchers. Warren [4] observed increased publication rates among newly hired social science scholars, finding that publishing expectations are now twice as great as they were in the early 1990's for graduates seeking an assistant professor position or assistant professors seeking promotion to associate professor. In a prior study, Bauldry [5] examined 403 new hires in sociology at 98 research universities and found assistant professors hired in either 2011 or 2012 had a median number of publications two to three times greater than new assistant professors hired in 2007. Increasing rates of co-authorship of social science journal articles have also been studied. In a study of 56 subject categories in the Web of Science: Social Science Citation Index, Henriksen [6] observed that larger increases in the share of co-authorships occur in disciplines using quantitative research methods, large data sets, and team research, as well as those with greater research specialization.

Among the suggestions that Kyvik and Aksnes [2] offered as contributing factors to the growth of scholarly publishing was the improvement of research environments and external funding. However, they focused primarily on the impact of external funding, simply noting that research environments had benefitted from the introduction of personal computers, databases, and the internet. In 2009, Lazer et al. [7] described the emergent field of computational social science "that leverages the capacity to collect and analyze data with an unprecedented breadth and depth and scale." King [8] describes the dramatic methodological changes in computational social sciences as "from isolated scholars toiling away on their own to larger scale, collaborative, interdisciplinary, lab-style research teams; and from a purely academic

pursuit to having a major impact on the world." Since those early days, data repositories such as the Inter-university Consortium for Political and Social Research (ICPSR) and the Harvard-MIT Data Center now make large datasets and technical services available to researchers. On-campus resources for social science researchers at American universities have become widespread. For example, every member institution of the Association of American Universities (AAU) now has at least one center, institute, or program exploring computational social science research. The availability of large research grants further demonstrates the growing importance of quantitative social sciences. The U.S. National Science Foundation (NSF) Directorate of Social, Behavioral and Economic Science, for example, funds projects through a program called *Human Networks and Data Science* [9]. The emergence and expansion of quantitative social sciences has had a clear impact on the social science research environment. Assembly and analysis of massive databases, data verification, statistical modeling, and visualization require levels of expertise beyond a single researcher. Jones' [10] recent analysis of all published articles in economics from 1950–2018 estimates that single-author articles ceased to be the majority of economics papers in 2005, and that co-authored papers now constitute 74% of all articles in the discipline.

Another potential factor in the growth of publishing offered by Kyvik and Aksnes [2] is the emergence and spread of performance-based research funding systems in which published output has become an important parameter in evaluation of individual scholars, their departments, and their universities. Hermanowicz [11] observed that both the university and the individual faculty member have become entrepreneurs for whom "research and publication have become the main currency in which prestige is traded." In short, research evaluation and the conferral of prestige share the same currency: scholarly publication. The UK's Research Excellence Framework (REF), for example, determines a large proportion of national funding for institutional research in the United Kingdom [12]. Fry et al. [13] conducted interviews in December, 2008 aimed at understanding how research assessment may influence scholarly and scientific publication in the UK. They reported a near-universal view among respondents that the publication of peer-reviewed journal articles was a fundamental disciplinary and institutional expectation, and that there was increasing institutional pressure to publish more frequently. Across disciplines, institutional emphasis was placed on peer-reviewed journal articles as the preferred output which would most contribute to their institution's REF submissions. Additionally, emphasis was placed on collaborative research, suggesting that collaborative team projects were best-suited to REF submissions. In their review of the evolution of UK economics under the REF, Lee et al. [14] note that over the course of four research assessment exercises, 1992, 1996, 2001, and 2008, the proportion of all journal submissions appearing in Diamond's [15] 27 core prestigious economics journals increased from 31% to 44%. Further, the percentage of journal titles in all economics departmental submissions increased from 53% in 1992 to 91% in 2008 [14].

Another analysis of REF submissions for the 1996, 2001, and 2008 REF cycles found that the volume of articles grew from 62% of submitted publications in 1996 to 75% in 2008 [16]. The increase in articles came at the expense of other publishing types: engineering submissions included fewer conference proceedings and social sciences submissions included fewer books. Evidence from the REF demonstrates that performance-based evaluation can catalyze more collaborative research and more frequent journal publication to the exclusion of other publication types. The United States does not have a national assessment framework tied to research funding, but widely consulted research evaluation data sources in the US also favor journal articles over books; the most recent National Research Council report on US doctoral programs, for example, did not include books in its tally of social science publications [17].

Influential public university rankings also omit books (e.g., QS World University Rankings) or minimize books' weight relative to journal articles (e.g., US News and World Report) [18, 19].

## Materials and methods

### Data source

We mined the Academic Analytics, LLC (AcA) commercial database for the names, departmental affiliation(s), and year of terminal degree of tenured and tenure-track scholars (Assistant Professor, Associate Professor, and Professor titles) over 9 years (2011–2019) in the following 12 social and behavioral science fields at Ph.D. granting universities in the United States:

1. Anthropology

2. Criminal Justice and Criminology

3. Economics

4. Educational Psychology

5. Geography

6. International Affairs and Development

7. Political Science

8. Psychology

9. Public Administration

10. Public Policy

11. Social Work/Social Welfare

12. Sociology

The AcA database compiles information on faculty members associated with academic departments at 380 Ph.D.-granting universities in the United States. AcA faculty rosters are updated at least annually by manual collection from publicly available resources, supplemented by verification and submission of faculty lists from some institutions. Each academic department is manually assigned to one or more of 170 subject classifications based on the National Center for Education Statistics (NCES) Classification of Instructional Programs (CIP) code classifications [20]. A complete list of the departments included in this study and their subject classifications is publicly available (https://osf.io/2x4uf/). AcA matches scholarly publications to their authors using a semi-automated matching process. All journal articles indexed in CrossRef (https://www.crossref.org/) are ingested into AcA's data warehouse and matched to their author(s); our study includes only the peer-reviewed journal articles, other article types that are also assigned DOIs but do not necessarily represent original research are excluded (e.g., book reviews, obituaries). Harzing [21] found that CrossRef has "similar or better coverage" of publications than Web of Science and Scopus, but are less comprehensive than Google Scholar and Microsoft Academic. A CrossRef API query performed in January 2022 reveals 6,200,221 works of all types are indexed in CrossRef with a publication date in 2019. Of these works, AcA identified 367,883 unique peer-reviewed journal articles (co-) authored by faculty members at the Ph.D. granting universities in their database (i.e., in 2019, about 5.9% of the works indexed in CrossRef represent peer-reviewed journal articles by authors at the institutions covered by AcA). Works indexed by CrossRef that were not

matched to scholars in the AcA database are either non-journal article types (e.g., conference proceedings, book chapters, working papers), or they were authored by scholars outside the United States or at non-Ph.D. granting US universities.

AcA also matches academic book publications from Baker & Taylor (https://www.baker-taylor.com/) to their respective author(s), editor(s), and translator(s). Baker & Taylor is among the most widely used book vendors among public libraries; scholarly books from 5,774 publishers catalogued by Baker & Taylor are matched to faculty members in the Academic Analytics database (the list of academic publishers is available at https://osf.io/2x4uf/). For both publication types, a 5-year window of authored publications was extracted (e.g., for the 2011 database, publications authored between 2007–2011 were extracted). All faculty members in departments assigned to one of the 12 social sciences disciplines were included in the sample, including those with zero articles or books in the previous 5 years.

The earliest iteration of the AcA database we extracted (2011) contains 27,447 unique faculty members affiliated with 1,476 social science departments at 267 universities that offered a social science Ph.D. degree in 2011. The most recent database (2019) contains 28,928 unique faculty members affiliated with 1,561 departments at 290 universities that offered a social science Ph.D. degree in 2019. Anonymized raw data, including faculty and department lists for each of the nine years studied, journal titles, book presses, and the crosswalk of university departments to scholarly disciplines are publicly available (https://osf.io/2x4uf/).

## Data analysis

All post-extract data handling, computations and statistical tests were performed in R v1.4.0 [22]. The total publication output of each discipline in each database year was tabulated as the unique number of articles and books published by scholars whose academic departments are classified within that discipline category. Each publication is counted only once per discipline, even if more than one scholar in that discipline shared authorship of that work. For each discipline and each year, we calculated the number of articles per faculty, the number of books per faculty, and the number of books per article. Changes in the number of departments in each discipline over the 9-year period may reflect the creation of new departments at the universities studied, but it may also reflect an increased scope of data collection in the AcA database. We attempted to control for the creation (or dissolution) of departments (and the possibility of increased faculty roster collection efforts) by calculating the same totals and ratios as above for the subset of departments that appear in the AcA database in all nine database years. Likewise, to explore whether changes in article and book publication are related to changing demographics within disciplines or due to changes in the publication practices of individual scholars, we calculated the same totals and ratios as above for the subset of faculty members who appear in all nine database years. The median number of authors on each article was also tabulated for each discipline in each year, to explore the growth of team authorship.

In each database year, the proportion of each discipline's population actively engaged in a particular publication type was calculated as the percent of scholars who published at least one book in the previous five years, and the percent who published at least one journal article in the previous five years. Significant differences in the proportion of scholars who have published at least one book (or journal article) between the 2011 and 2019 years was tested using the Chi-squared test.

The AcA database includes the year of terminal degree for each faculty member (typically the Ph.D., but sometimes MBA, MFA, etc.), from which we defined three academic age cohorts following [23]: early career researchers (ECR) earned their terminal degree 0–10 years before the year in which the database was compiled; mid-career researchers (MCR) earned their

degree between 11 and 30 years before the database compilation year; and senior career researchers (SCR) earned their degree 31 or more years before the database compilation year. For each discipline, year, and age cohort we calculated the number of articles per faculty, the number of books per faculty, and the number of books per article. The proportion of each age cohort participating in both publication types was also calculated, and differences in the percent of the population actively engaged in each publication type was tested using the Chi-squared test. When comparisons are made with disciplines among age cohorts, the unique number of books or articles authored by that age cohort was used.

## Results

### Population, academic department count, and publication count

The number of faculty members and academic departments in each discipline in the 2011 and 2019 database years, as well as the percent change between 2011 and 2019, appears in Table 1. Data for all years is available at https://osf.io/2x4uf/. In total, scholars in the 2011 dataset published 158,104 unique journal articles in 8,706 journals between 2007 and 2011. Over the same five-year period, the 2011 scholars published 17,101 unique books. Scholars in the 2019 dataset published 215,540 unique journal articles in 11,480 journals between 2015 and 2019. Over the same five-year period, the 2019 scholars published 13,102 unique books. The counts of unique journal articles and books by authors in each discipline are presented in Table 2. While the overall number of social science scholars increased 5.4% between 2011 and 2019, the number of journal articles they produced increased at a much faster rate: 36.3%. The increase in articles published is associated with a 31.9% increase in the number of unique journal titles in which these works appear. Conversely, the overall number of books published decreased by 23.4% over the nine-year period. The declining ratio of books per journal article in each discipline over the study period is shown in Fig 1.

Growth in the number of social science faculty members over the study period (5.4%) is not uniform across disciplines. Table 1 reveals that one discipline (Educational Psychology) evinced a slight decline in the number of faculty members and the number of academic departments (-1.2%), while large population increases were observed in International Affairs and Development (39.1%), Public Administration (22.3%), and Public Policy (10.1%). The number of academic departments classified as International Affairs and Development and Public Administration also increased substantially between 2011 and 2019 (Table 1). A different pattern is observed among Criminal Justice and Criminology departments. The number of departments in this discipline increased by more than 10% over the study period, while the

**Table 1. Faculty member population and number of academic departments in the 2011 and 2019 database snapshots.**

| Discipline | Faculty 2011 | Faculty 2019 | % Change Faculty | Departments 2011 | Departments 2019 | % Change Depts |
|---|---|---|---|---|---|---|
| Anthropology | 2,788 | 2,886 | 3.5% | 180 | 189 | 5.0% |
| Criminal Justice and Criminology | 1,081 | 1,139 | 5.4% | 78 | 86 | 10.3% |
| Economics | 4,318 | 4,492 | 4.0% | 213 | 223 | 4.7% |
| Educational Psychology | 1,148 | 1,134 | -1.2% | 63 | 61 | -3.2% |
| Geography | 1,502 | 1,558 | 3.7% | 103 | 102 | -1.0% |
| International Affairs and Development | 1,030 | 1,433 | 39.1% | 40 | 52 | 30.0% |
| Political Science | 4,233 | 4,295 | 1.5% | 201 | 210 | 4.5% |
| Psychology | 6,087 | 6,537 | 7.4% | 248 | 267 | 7.7% |
| Public Administration | 1,006 | 1,230 | 22.3% | 59 | 68 | 15.3% |
| Public Policy | 1,650 | 1,816 | 10.1% | 80 | 81 | 1.2% |
| Social Work/Social Welfare | 2,104 | 2,269 | 7.8% | 127 | 138 | 8.7% |
| Sociology | 3,300 | 3,398 | 3.0% | 195 | 207 | 6.2% |

**Table 2. Faculty, article (5-year count), and book (5-year count) totals and ratios in 2011 and 2019.**

| Discipline | Year | Faculty Count | Article Count (5yr) | Articles (5yr) per Person | Articles per person % change | Book Count (5yr) | Books (5yr) per Person | Books per person % change | Books per Article (5yr) | Books per article % change |
|---|---|---|---|---|---|---|---|---|---|---|
| Anthropology | 2011 | 2,788 | 11,486 | 4.12 | | 2,108 | 0.76 | | 0.18 | |
| | 2019 | 2,886 | 15,939 | 5.52 | 34% | 1,721 | 0.60 | -21% | 0.11 | -41% |
| Criminal Justice and Criminology | 2011 | 1,081 | 5,095 | 4.71 | | 901 | 0.83 | | 0.18 | |
| | 2019 | 1,139 | 7,427 | 6.52 | 38% | 684 | 0.60 | -28% | 0.09 | -48% |
| Economics | 2011 | 4,318 | 22,715 | 5.26 | | 1,982 | 0.46 | | 0.09 | |
| | 2019 | 4,492 | 24,310 | 5.41 | 3% | 1,117 | 0.25 | -46% | 0.05 | -47% |
| Educational Psychology | 2011 | 1,148 | 6,098 | 5.31 | | 540 | 0.47 | | 0.09 | |
| | 2019 | 1,134 | 9,852 | 8.69 | 64% | 460 | 0.41 | -14% | 0.05 | -47% |
| Geography | 2011 | 1,502 | 10,277 | 6.84 | | 767 | 0.51 | | 0.07 | |
| | 2019 | 1,558 | 16,395 | 10.52 | 54% | 560 | 0.36 | -30% | 0.03 | -54% |
| International Affairs and Development | 2011 | 1,030 | 4,797 | 4.66 | | 1,266 | 1.23 | | 0.26 | |
| | 2019 | 1,433 | 7,403 | 5.17 | 11% | 1,266 | 0.88 | -28% | 0.17 | -35% |
| Political Science | 2011 | 4,233 | 15,918 | 3.76 | | 4,128 | 0.98 | | 0.26 | |
| | 2019 | 4,295 | 19,861 | 4.62 | 23% | 3,221 | 0.75 | -23% | 0.16 | -37% |
| Psychology | 2011 | 6,087 | 59,739 | 9.81 | | 2,541 | 0.42 | | 0.04 | |
| | 2019 | 6,537 | 85,755 | 13.12 | 34% | 1,894 | 0.29 | -31% | 0.02 | -48% |
| Public Administration | 2011 | 1,006 | 5,238 | 5.21 | | 686 | 0.68 | | 0.13 | |
| | 2019 | 1,230 | 8,084 | 6.57 | 26% | 735 | 0.60 | -12% | 0.09 | -31% |
| Public Policy | 2011 | 1,650 | 11,644 | 7.06 | | 1,236 | 0.75 | | 0.11 | |
| | 2019 | 1,816 | 15,760 | 8.68 | 23% | 1,066 | 0.59 | -22% | 0.07 | -36% |
| Social Work/Social Welfare | 2011 | 2,104 | 9,827 | 4.67 | | 830 | 0.39 | | 0.08 | |
| | 2019 | 2,269 | 17,012 | 7.50 | 61% | 655 | 0.29 | -27% | 0.04 | -54% |
| Sociology | 2011 | 3,300 | 14,767 | 4.47 | | 2,428 | 0.74 | | 0.16 | |
| | 2019 | 3,398 | 20,544 | 6.05 | 35% | 1,930 | 0.57 | -23% | 0.09 | -43% |

number of faculty members in this field increased by 5.4% on par with the overall social sciences population increase.

The results presented in Table 2 are not substantively different than those performed on a subset of the data limited to only those departments appearing in all nine years of the dataset. The limited subset of departments differed from the complete set by about 1%, on average, in terms of articles per person, books per person, and books per article. A table showing results for the limited subset of departments is available at https://osf.io/2x4uf/. Similarity between the full sample of departments and the limited subset suggest that growth in the number of departments (or expanded AcA data collection efforts) does not account for the trend towards increased article publication and decreased book publication we observed in each discipline.

## Changing publishing practices

In every discipline, the 5-year total of journal article publications increased more rapidly than the population of faculty members, and in all but one discipline the 5-year total of book publications decreased between 2011 and 2019 (in Public Administration, book publishing increased by 7.1%). Book publishing in International Affairs and Development also increased

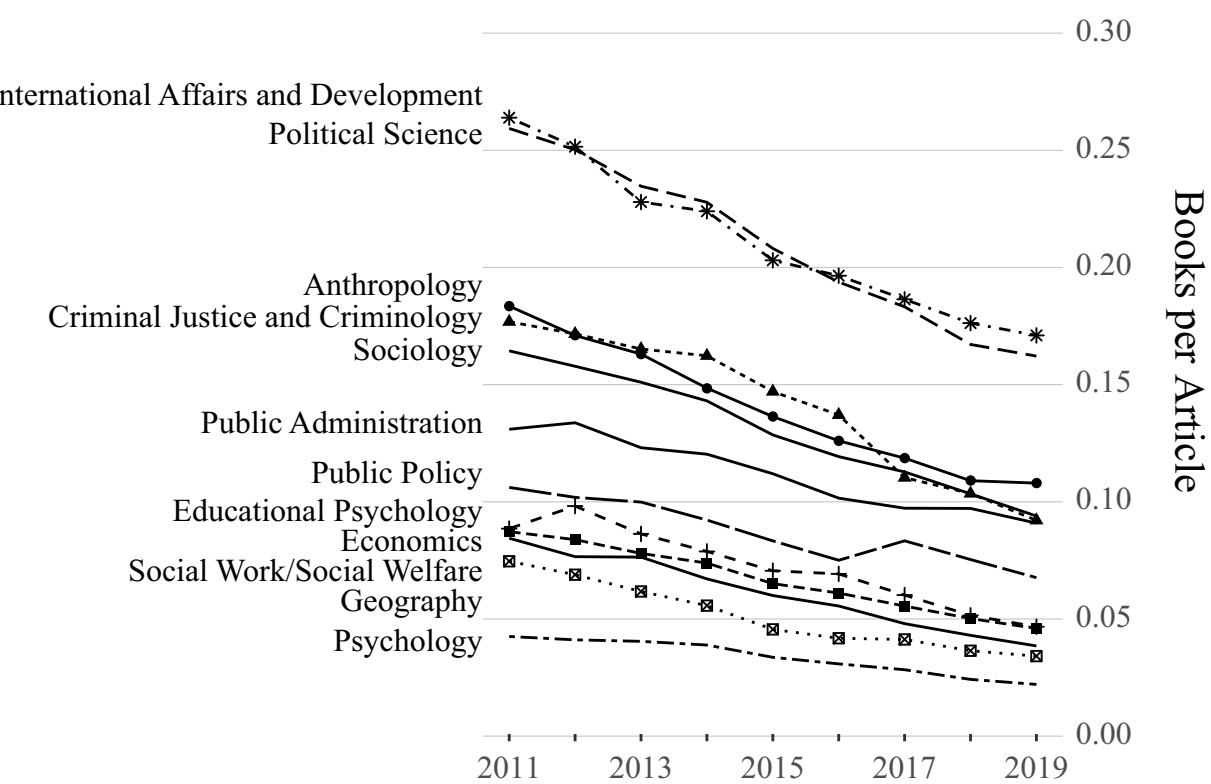

**Fig 1. Books per journal article between 2011 and 2019.** Both books and journal articles represent five-year sums, such that the 2011 data point represents the total books published between 2007 and 2011 divided by the total articles published between 2007 and 2011, the 2012 data point represents the total books published between 2008 and 2012 divided by the total articles published between 2008 and 2012, and so on.

after 2011 but began to decrease again after 2014 and by 2019 the number of books published was identical to the 2011 number.

Table 2 shows the number of journal articles and books in 2011 and 2019 for each discipline, as well as the following ratios: articles per faculty, books per faculty, and books per article. In seven of the 12 disciplines we studied, journal articles per person increased by more than 30% from 2011 to 2019, including a more than 60% increase in both Educational Psychology and Social Work/Social Welfare. The increase in articles per person in Economics is notably lower than the other disciplines, with only a 3% increase over the study period. The next lowest increase is in International Affairs and Development (11%); unlike other fields, the year-by-year population growth in International Affairs and Development largely mirrors growth in the discipline's journal article output over time.

A decline in books per person over the study period characterizes all twelve disciplines (Table 2). Economics shows the greatest decline in books per person (-46%), while the lowest declines are in Public Administration and Educational Psychology (-12% and -14%, respectively). In all disciplines, the ratio of books per article declined by at least 31% from 2011 to 2019. In Sociology, for example, there was one book published for every 6.3 journal articles in the 2011 dataset; in the 2019 dataset, there was one book published for every 11.1 articles. The largest decline in books per article appears in Geography and Social Work/Social Welfare (-54% in both disciplines). We visualized the ratio "books per journal article" in each discipline throughout the study period (Fig 1); book publications constitute a steadily decreasing portion of the total publication output in each social science discipline over this timeframe.

**Table 3. Article (5-year count), and book (5-year count) per person ratios in 2011 and 2019 for each age cohort.**

| Discipline | Age cohort | 2011 articles per person (5yr) | 2019 articles per person (5yr) | Articles per person % change | 2011 books per person (5yr) | 2019 books per person (5yr) | Books per person % change |
|---|---|---|---|---|---|---|---|
| Anthropology | ECR | 4.40 | 6.49 | 47.4% | 0.41 | 0.38 | -9.0% |
| | MCR | 4.52 | 5.98 | 32.4% | 0.85 | 0.57 | -32.6% |
| | SCR | 3.95 | 5.32 | 34.9% | 1.01 | 0.93 | -7.7% |
| Criminal Justice and Criminology | ECR | 5.49 | 8.72 | 58.7% | 0.48 | 0.25 | -48.2% |
| | MCR | 5.33 | 6.85 | 28.6% | 0.87 | 0.73 | -15.8% |
| | SCR | 4.29 | 6.94 | 61.6% | 1.15 | 1.03 | -10.8% |
| Economics | ECR | 4.64 | 4.65 | 0.3% | 0.04 | 0.03 | -32.0% |
| | MCR | 6.47 | 6.56 | 1.5% | 0.43 | 0.21 | -50.2% |
| | SCR | 5.49 | 6.53 | 19.0% | 0.99 | 0.58 | -42.0% |
| Educational Psychology | ECR | 5.78 | 10.31 | 78.4% | 0.17 | 0.15 | -12.6% |
| | MCR | 5.99 | 9.03 | 50.7% | 0.48 | 0.55 | 15.8% |
| | SCR | 5.48 | 8.07 | 47.3% | 1.01 | 0.50 | -51.0% |
| Geography | ECR | 6.75 | 9.97 | 47.7% | 0.17 | 0.20 | 16.1% |
| | MCR | 8.20 | 12.45 | 51.7% | 0.62 | 0.38 | -39.3% |
| | SCR | 6.22 | 10.88 | 74.8% | 0.87 | 0.65 | -25.6% |
| International Affairs and Development | ECR | 4.35 | 5.64 | 29.6% | 0.53 | 0.46 | -13.8% |
| | MCR | 4.84 | 5.20 | 7.5% | 1.19 | 0.88 | -26.0% |
| | SCR | 4.96 | 5.14 | 3.6% | 2.07 | 1.35 | -34.8% |
| Political Science | ECR | 4.04 | 5.51 | 36.5% | 0.47 | 0.41 | -12.8% |
| | MCR | 4.15 | 4.89 | 17.8% | 1.01 | 0.76 | -24.8% |
| | SCR | 3.77 | 4.51 | 19.8% | 1.63 | 1.25 | -23.4% |
| Psychology | ECR | 10.38 | 14.26 | 37.3% | 0.11 | 0.06 | -45.3% |
| | MCR | 11.94 | 15.65 | 31.1% | 0.41 | 0.29 | -29.1% |
| | SCR | 11.13 | 15.34 | 37.8% | 0.75 | 0.52 | -30.5% |
| Public Administration | ECR | 5.48 | 7.42 | 35.5% | 0.26 | 0.21 | -18.4% |
| | MCR | 5.24 | 6.54 | 24.8% | 0.68 | 0.63 | -6.8% |
| | SCR | 5.89 | 7.23 | 22.7% | 1.29 | 1.14 | -11.7% |
| Public Policy | ECR | 6.27 | 8.33 | 32.8% | 0.27 | 0.19 | -29.1% |
| | MCR | 7.87 | 9.69 | 23.1% | 0.75 | 0.55 | -26.7% |
| | SCR | 7.97 | 9.02 | 13.2% | 1.22 | 1.10 | -9.9% |
| Social Work/Social Welfare | ECR | 5.58 | 9.76 | 74.8% | 0.14 | 0.09 | -33.6% |
| | MCR | 5.15 | 8.31 | 61.4% | 0.45 | 0.35 | -23.0% |
| | SCR | 4.88 | 7.86 | 60.9% | 0.81 | 0.66 | -18.9% |
| Sociology | ECR | 4.79 | 7.01 | 46.4% | 0.35 | 0.32 | -8.8% |
| | MCR | 5.02 | 6.50 | 29.4% | 0.80 | 0.58 | -26.9% |
| | SCR | 4.70 | 6.26 | 33.4% | 1.18 | 0.93 | -20.8% |

Table 3 shows 2011 and 2019 articles per person and books per person by age cohort. In eight of twelve disciplines, the largest increase in articles per person from 2011 to 2019 is observed among the youngest age cohort (ECR). ECRs in Economics showed the smallest increase in journal articles per person among the ECR cohorts. Book publications per person declined among SCRs in all disciplines, and books per person also declined among MCRs in all but one discipline (Educational Psychology). Books per person among ECRs decreased slightly in all but one discipline, Geography.

Analysis of a subset of data containing only individual faculty members that appear in all nine years of the study is available at https://osf.io/2x4uf/. Data show that the number of journal articles authored by faculty who were present throughout the study period did increase in most disciplines, but this increase was much less than that observed for the overall study sample (e.g., faculty present throughout the study timeframe in Anthropology authored 15.4% more articles in 2019 than in 2011, but when the entire sample of faculty members is included, the increase was more than twice as great, 34.1%). This result indicates that while all faculty members are contributing to the increase in journal article production, those who joined the faculty at the research institutions in our study after 2011 (i.e., new hires) contributed disproportionately to the overall increase in journal article authorships over the study period.

## Percent of faculty actively engaged in publishing books and articles

We quantified the rate of participation in each publishing type in each of the nine years studied. Results for 2011 and 2019 appear in Table 4 along with significance values from Chi-squared tests for differences in proportions of faculty who participate in a particular publishing type; results for each of the nine years are available at https://osf.io/2x4uf/. The rate of participation was defined as the percent of all scholars in each discipline within a data year who authored at least one of that type of publication over the previous 5-year period, divided by the total number of scholars in that discipline in that year. In every discipline, the rate of participation in journal article publishing increased, while the rate of book publishing participation decreased. The changes in participation rate between 2011 and 2019 observed in Table 4 are generally less than the changes in books and articles per person (Table 2). For example, although Geography saw a 54% decline in the number of books published per person, the number of faculty who have published at least one book decreased by only 5.9%. Likewise, in Psychology the number of journal articles per person increased 34% over the study period, but the percent of the population engaged in the production of journal articles increased only 2.9%.

Table 5 shows the percent of scholars in 2011 and 2019 who published at least one book or journal article by age cohort. In all but one discipline (Economics) the youngest cohort (ECR)

**Table 4. Percent of the population of scholars who have published at least one journal article or book in the 5-year period preceding in 2011 and 2019.**

| Discipline | 2011% scholars with an article | 2019% scholars with an article | 2011–2019 change in article participation | $p < 0.05$ | 2011% scholars with a book | 2019% scholars with a book | 2011–2019 change in book participation | $p < 0.05$ |
|---|---|---|---|---|---|---|---|---|
| Anthropology | 83.1% | 87.9% | 4.8% | * | 42.3% | 37.5% | -4.8% | * |
| Criminal Justice and Criminology | 79.3% | 86.7% | 7.5% | * | 38.6% | 29.7% | -8.9% | * |
| Economics | 83.0% | 87.5% | 4.6% | * | 19.4% | 12.5% | -6.9% | * |
| Educational Psychology | 84.9% | 90.7% | 5.8% | * | 26.7% | 24.3% | -2.5% | |
| Geography | 88.5% | 92.7% | 4.2% | * | 27.7% | 21.8% | -5.9% | * |
| International Affairs and Development | 81.0% | 83.7% | 2.8% | | 51.2% | 44.8% | -6.4% | * |
| Political Science | 81.3% | 86.0% | 4.8% | * | 47.6% | 42.6% | -5.0% | * |
| Psychology | 90.4% | 93.3% | 2.9% | * | 22.0% | 17.0% | -4.9% | * |
| Public Administration | 79.1% | 85.3% | 6.2% | * | 32.5% | 29.5% | -3.0% | |
| Public Policy | 83.5% | 87.0% | 3.5% | * | 34.0% | 29.6% | -4.4% | * |
| Social Work/Social Welfare | 79.3% | 87.7% | 8.3% | * | 22.3% | 17.5% | -4.8% | * |
| Sociology | 84.8% | 88.3% | 3.5% | * | 36.8% | 32.9% | -3.8% | * |

P-value is from Chi-squared test for difference in proportions who have published in the two years.

**Table 5. Percent of the population of scholars by age cohort who have published at least one journal article or book in the 5-year period preceding 2011 and 2019.**

| Discipline | Age cohort | 2011% of scholars with an article | 2019% of scholars with an article | 2011–2019 change in article participation | $p < 0.05$ | 2011% of scholars with a book | 2019% of scholars with a book | 2011–2019 change in book participation | $p < 0.05$ |
|---|---|---|---|---|---|---|---|---|---|
| Anthropology | ECR | 88.8% | 94.5% | 5.7% | * | 31.7% | 29.6% | -2.0% | |
| | MCR | 84.0% | 87.5% | 3.5% | * | 46.5% | 37.2% | -9.4% | * |
| | SCR | 72.6% | 80.4% | 7.8% | * | 48.5% | 48.1% | -0.4% | |
| Criminal Justice and Criminology | ECR | 88.3% | 95.7% | 7.5% | * | 28.2% | 17.3% | -11.0% | * |
| | MCR | 80.2% | 84.8% | 4.6% | | 43.5% | 35.4% | -8.1% | * |
| | SCR | 64.2% | 71.2% | 7.0% | | 44.0% | 41.8% | -2.2% | |
| Economics | ECR | 83.6% | 87.5% | 3.9% | * | 3.7% | 2.0% | -1.7% | * |
| | MCR | 87.3% | 91.8% | 4.5% | * | 21.6% | 11.5% | -10.1% | * |
| | SCR | 74.6% | 80.7% | 6.1% | * | 35.6% | 27.5% | -8.1% | * |
| Educational Psychology | ECR | 88.4% | 93.8% | 5.4% | * | 13.9% | 10.6% | -3.4% | |
| | MCR | 84.4% | 89.5% | 5.1% | * | 29.7% | 31.8% | 2.1% | |
| | SCR | 79.8% | 87.9% | 8.1% | * | 42.4% | 29.7% | -12.7% | * |
| Geography | ECR | 92.9% | 96.3% | 3.3% | * | 12.1% | 11.8% | -0.3% | |
| | MCR | 90.4% | 91.9% | 1.5% | | 33.4% | 24.5% | -8.9% | * |
| | SCR | 73.8% | 88.5% | 14.7% | * | 41.4% | 32.3% | -9.1% | * |
| International Affairs and Development | ECR | 87.1% | 91.7% | 4.7% | * | 33.0% | 32.8% | -0.2% | |
| | MCR | 81.7% | 86.2% | 4.5% | * | 57.6% | 47.9% | -9.7% | * |
| | SCR | 73.0% | 70.5% | -2.6% | | 59.9% | 51.8% | -8.1% | * |
| Political Science | ECR | 88.0% | 93.8% | 5.7% | * | 34.7% | 31.2% | -3.5% | |
| | MCR | 82.7% | 85.8% | 3.1% | * | 52.7% | 44.6% | -8.2% | * |
| | SCR | 68.3% | 74.5% | 6.2% | * | 57.0% | 55.5% | -1.6% | |
| Psychology | ECR | 95.4% | 97.7% | 2.3% | * | 9.3% | 4.8% | -4.5% | * |
| | MCR | 92.1% | 93.8% | 1.8% | * | 23.3% | 17.6% | -5.7% | * |
| | SCR | 81.7% | 87.5% | 5.8% | * | 33.0% | 28.4% | -4.6% | * |
| Public Administration | ECR | 87.1% | 94.2% | 7.1% | * | 19.4% | 14.2% | -5.1% | |
| | MCR | 76.5% | 85.0% | 8.4% | * | 37.0% | 34.1% | -2.8% | |
| | SCR | 72.4% | 72.0% | -0.4% | | 43.2% | 42.8% | -0.4% | |
| Public Policy | ECR | 89.1% | 93.9% | 4.8% | * | 20.0% | 13.7% | -6.3% | * |
| | MCR | 84.3% | 88.5% | 4.2% | * | 35.9% | 31.9% | -4.0% | |
| | SCR | 76.6% | 76.4% | -0.2% | | 45.5% | 44.1% | -1.3% | |
| Social Work/Social Welfare | ECR | 87.4% | 93.7% | 6.4% | * | 11.5% | 6.7% | -4.7% | * |
| | MCR | 77.0% | 85.7% | 8.7% | * | 26.3% | 21.6% | -4.7% | * |
| | SCR | 66.8% | 77.4% | 10.6% | * | 36.0% | 34.1% | -1.9% | |
| Sociology | ECR | 91.1% | 95.6% | 4.5% | * | 25.7% | 23.2% | -2.5% | |
| | MCR | 85.2% | 88.4% | 3.1% | * | 40.2% | 34.1% | -6.1% | * |
| | SCR | 74.9% | 76.4% | 1.5% | | 46.4% | 45.6% | -0.8% | |

P-value is from Chi-squared test for difference in proportions who have published in the two years.

ECR = early career researcher, 0–10 years since terminal degree. MCR = mid-career researcher, 11–30 years since terminal degree. SCR = senior career researcher, 31 or more years since terminal degree.

had the greatest rate of participation in journal article publication in both 2011 and 2019. Likewise, in every discipline except Educational Psychology, the oldest cohort (SCR) had the greatest participation in book publication. These findings are broadly consistent with our previous analysis of publishing behavior among age cohorts [23: Fig 6], where senior scholars were observed to publish more books than their younger colleagues. Table 5 also shows that rates of participation in journal article publication increased in all age cohorts in all disciplines, with three exceptions: SCRs in International Affairs and Development, Public Administration, and Public Policy all showed non-significant decreases in journal article publication participation. In six disciplines SCRs showed the greatest increase in journal article publication participation, in four disciplines the greatest increase was among ECRs, and in the remaining two disciplines MCRs showed the greatest increase in journal article publication participation.

Thirty-five of the 36 comparisons of book publication participation by age cohort revealed a decrease in participation rate between 2011 and 2019 (among MCRs in Educational Psychology the increase was not statistically significant; Table 5). In six disciplines MCRs showed the greatest decrease in journal article publication participation, in four disciplines the greatest decrease was among ECRs, and in the remaining two disciplines SCRs showed the greatest increase in journal article publication participation. ECRs universally show the lowest rate of book publication participation. This is most extreme in Economics where only 3.7% of ECRs published at least one book in the 5-year period leading to 2011, and only 2.0% of Economics ECRs published a book in the 5-year period leading to 2019.

## Discussion

Individual scholars are members of various communities: academic departments, colleges, universities, and disciplines, among others. Placing the individual author in this complex social context, Nygaard [24] used an academic literacies framework to analyze research and writing. In this model, research, writing, and publishing are social practices embedded within a community, and communities create expectations for individual behavior. The researcher must decide the genre of the artifact to be produced, whether to involve other researchers in a collaborative effort, the quality of the work, the appropriate audience, and the process of how the scholarship is done. The community (department, university, discipline, etc.) establishes the parameters for those individual decisions. One of the most consequential decisions early career faculty face is deciding the venue for publishing their research. Clemens et al. [25] observe that access to book publishers is usually through cumulative advantage which accrues to senior faculty who have established a record of successful publications. Journal article publication, on the other hand, is more egalitarian, relying more on the author's tenacity to submit their work multiple times until accepted. Thus, as Harley, et al. [26], Tenopir, et al. [27], and Wakeling, et al. [28] suggest, early career faculty members often recognize that the most advantageous strategy is to first establish their research reputations through the publication of journal articles in prestigious journals. With this background to the decisions the publishing researcher makes and the choices available, we suggest that the growing pressure to publish more—and more frequently—amidst the backdrop of community, reputation, and career stage requirements has altered the publication practices of social scientists.

Journal articles are the *de facto* "currency" of research in many physical, mathematical, biological, biomedical, and engineering fields [e.g., 29], and our data show that the social sciences are becoming more like those STEM disciplines in terms of publication practices. King [8] prefaced his comments on how computational research is restructuring the social sciences by noting "The social sciences are in the midst of an historic change, with large parts moving from the humanities to the sciences in terms of research style, infrastructural needs, data

availability, empirical methods, substantive understanding, and the ability to make swift and dramatic progress." Thus, the research methodologies of large parts of the social sciences are contributing to more collaborative research and an increased emphasis on journal article publication. Our analyses suggest that the emphasis on journal article publication may come at the expense of book publication and may be driven by increasing article publishing expectations on the youngest age cohort. While increased rates of journal article publication are not limited to the ECR cohort, in all but one discipline, the youngest cohort (ECR) had the greatest rate of participation in journal article publication in both 2011 and 2019. Our finding that the increase in articles per faculty member among those who appear in all nine years of the study is less than the overall increase in article per author may be partially explained by increasing pressure to write more papers among ECRs, perhaps as a corollary to the increasingly competitive job market for professorships.

The influence of performance-based research assessment systems on faculty publishing and research decisions is also likely related to the increase in journal article production and the de-emphasis on book publication. Hicks [30] notes ". . ..it is the form of social science scholarly publication that is evolving in response to the imposition of national research evaluation. . . Research evaluation and publishing in the social sciences and humanities are co-evolving." Our data indicate that this co-evolution in the social sciences likely results in greater emphasis on large research programs conducted by teams and increased frequency of journal article publication. In every discipline we examined, the rate of participation in journal article publishing increased, while the rate of book publishing participation decreased. In general, books take more time to produce than journal articles and their impact on the community is difficult to ascertain in the short term due in part to a dearth of comprehensive book citation databases. We posit that the increased need for rapidly produced research artifacts, the growth of quantitatively focused modes of inquiry in social science disciplines, and the increasingly greater number of journal articles produced by ECRs is likely to continue favoring journal article publication in the social sciences over book publications.

There are several potential ramifications of the decrease in book publications for social sciences as a whole and individual social science disciplines. The U.S. market for scholarly monographs has been shrinking for several years [31]. Book publishers used to see successful print runs and sales of 2,000 copies of new books. Now, annual sales of 200 copies of a new book is considered successful by some publishers [32]. Some book publishers have responded to this decline in revenues by increasing book prices as much as three-or four-fold [32]. The declines in book publications may provide some relief for acquisition librarians stretching their already depleted funds.

Declining book publications may have detrimental effects for social sciences disciplines most closely related to the humanities. Long-form scholarly publishing provides the place and space to explore a topic in detail, analyzing subjects with greater contextualization than shorter-form journal articles typically allow. Crossik [33] observes that "Writing a monograph allows the author to weave a complex and reflective narrative, tying together a body of research in a way that is not possible with journal articles or other shorter outputs." Hill [34] further suggests that "ways of knowing" and "forms of telling" are entwined; "reducing one may diminish the richness of the other."

## Future directions and study limitations

Collaborative research and publication have become commonplace in most disciplines in the social sciences, and further studies of scholarly collaboration are likely to provide more context and depth to understanding the behaviors involved in this phenomenon. Our study aimed to

quantify the disciplinary literature as a whole, rather than the number of authorships attributable to individual scholars. If an article was co-authored by more than one scholar within a discipline, the article was counted only once in the article total for that discipline. Articles per person (as reported, e.g., in Table 2) was calculated as the number of unique articles authored by scholars in that discipline, divided by the number of unique scholars in that discipline. In this way, our study design is not suited to directly address whether increased co-authorship has a causal relationship with increased journal article authorships overall. We did, however, calculate the median number of authors for each article in each calendar year between 2007 and 2019 (a table with these data appears as supplemental information (https://osf.io/2x4uf/). The median number of authors increased by 1.0 in all 12 disciplines we studied (e.g., from 2.0 to 3.0 authors per article in Anthropology, and from 3.0 to 4.0 authors per article in Geography). Increasing numbers of authors per article (see also [6, 10]), in light of our result that the number of unique articles per person is also increasing, suggests a fruitful avenue for future research to explore the relationship between "teaming" and disciplinary article production.

Our study sample was limited to research universities that offer the Ph.D. degree in the United States. Faculty employed by many non-Ph.D. granting universities also routinely publish research results, and are likely also influenced by community practices, prestige, external evaluation, and the increasing use of quantitative research methods. Future research may seek to expand the sample of universities to include those institutions. Further, we did not consider non-traditional forms of scientific and scholarly communication such as blog authorship, zine authorship, newsletters, op-ed pieces, listserv posts, performances, musical compositions, choreography, etc., nor did we consider conference proceedings and book chapters. It is possible that the decline we observed in books published (and the increase in journal articles published) does not completely capture the fullness of the shift in social science research dissemination strategies. Bibliometric data aggregation would benefit from the inclusion of more of the diversity of dissemination strategies now available to scholars.

## Acknowledgments

We thank the following individuals for thoughtful comments and suggestions: R. Berdahl, P. Lange, M. Matier, T. Stapleton, G. Walker, R. Wheeler, and C. Whitacre.

## Author Contributions

**Conceptualization:** William E. Savage, Anthony J. Olejniczak.

**Data curation:** William E. Savage, Anthony J. Olejniczak.

**Formal analysis:** Anthony J. Olejniczak.

**Investigation:** William E. Savage.

**Methodology:** William E. Savage, Anthony J. Olejniczak.

**Project administration:** Anthony J. Olejniczak.

**Visualization:** Anthony J. Olejniczak.

**Writing – original draft:** William E. Savage.

**Writing – review & editing:** William E. Savage, Anthony J. Olejniczak.

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
