## [Decision Letter · Decision Letter 0]

27 Oct 2021

PONE-D-21-27877More journal articles and fewer books: Publication practices in the social sciences in the 2010’sPLOS ONE

Dear Dr. Olejniczak,

Thank you for submitting your manuscript to PLOS ONE. After careful consideration, we feel that it has merit but does not fully meet PLOS ONE’s publication criteria as it currently stands. Therefore, we invite you to submit a revised version of the manuscript that addresses the points raised during the review process. The reviewers offer a number of useful suggestions for you to consider.  Among these I call your attention in particular to the following:All three reviewers ask  in their own way for you to clarify what is captured in the Academic Analytics data, addressing how these data are generated and clarifying any potential biases or mis-measurements that result from how the data are generated.Reviewer #2 offers a useful typology of possible sources of change that could be driving the results you document.  You should consider whether this framework is helpful, and the extent to which you can distinguish between these causes of change.Reviewers #1 and #3 suggest ways that you might exploit the panel nature of your data (repeated observations on the same individual at different dates) to help illuminate what is driving the changes you observe.Reviewer #3 suggests being clearer about what your research question is in the abstract and at the start of the article.  This reviewer also suggests reducing and tightening the literature review.

I encourage you to consider how these and the other points noted in the reviews might improve your article.

We look forward to receiving your revised manuscript.

Kind regards,

Joshua L Rosenbloom

Academic Editor

PLOS ONE

Journal Requirements:

“AJO and WES are paid employees of Academic Analytics, LLC. None of the authors has an equity interest in Academic Analytics, LLC. The results presented reflect the authors’ opinions, and do not necessarily reflect the opinions or positions of Academic Analytics, LLC. Academic Analytics, LLC management had no oversight or involvement in the project and were not involved in preparation or review of the manuscript. All work was done as part of the respective authors’ research, with no additional or external funding.”

Reviewers' comments:

Reviewer's Responses to Questions

**Comments to the Author**

1. Is the manuscript technically sound, and do the data support the conclusions?

Reviewer #1: Partly

Reviewer #2: Partly

Reviewer #3: Yes

2. Has the statistical analysis been performed appropriately and rigorously? 

Reviewer #1: Yes

Reviewer #2: Yes

Reviewer #3: Yes

3. Have the authors made all data underlying the findings in their manuscript fully available?

Reviewer #1: Yes

Reviewer #2: Yes

Reviewer #3: Yes

4. Is the manuscript presented in an intelligible fashion and written in standard English?

Reviewer #1: Yes

Reviewer #2: Yes

Reviewer #3: Yes

5. Review Comments to the Author

Reviewer #1: While I doubt the results of this study will be a surprise to most researchers in social science fields over the past decade, I have never seen data on these facts collected and analyzed before. To the best of my knowledge, the analysis is correct and, with two small caveats, meets the PLOS One standards for publication.

The first issue with the paper is that the Academic Analytics commercial database should be described in more detail. As this is a commercial database, it is unlikely to be familiar to most researchers, and without a little bit more detail it is difficult to put the results into context. In particular, should we think of the data as containing the universe of all faculty, only those faculty with a publication, only faculty from a subset of institutions, or only a sample of faculty? Are the same faculty included in multiple waves of the data, and if so, might a panel data structure allow the authors to say more about how an individuals’ publication behaviors are changing over time? The number of departments is rising over time (Table 1). Is this due to the creation of new academic departments, or to the Academic Analytics database obtaining more complete coverage over time? Answering these kinds of questions is important for interpreting the results.

Second, the authors make frequent references to new generations of researchers showing different publication patterns. Methodologically, the paper needs some discussion of the age-time-cohort issues: are we observing differences across generations, differences within individuals as they age, or a secular change over time. The fact that both junior and senior faculty publish more papers and fewer books would lead me to believe that this is a secular change in publication practices rather than a generational issue, although the effect is larger for younger researchers. In general, it is not possible to separately identify age versus cohort versus time effects, and so the authors should be even more careful when describing this set of results.

Reviewer #2: I appreciated having the opportunity to review the manuscript titled “More journal articles and fewer books: Publication practices in the social sciences in the 2010’s”. In the paper, the author(s) examine a structural shift in the social sciences whereby academics have concentrated more of their effort on journal publications rather than books. Using a relatively new and unique dataset, the authors are able to bring important new facts to bear about how scientific research is disseminated and rewarded. Although I enjoyed reading the manuscript and found the set of stylized facts quite interesting, I did find there to be room for improvement and additional clarification. Below, I have detailed some suggestions about how to address these concerns.

Major Points:

• An expanded discussion of the dataset is critical. Although the authors cite Academic Analytics as the source, it is unclear whether these data contain both the faculty rosters or publication data or both. Regardless, it seems critical to explain how these data are generated and how the rosters are linked with the bibliometric data. How do these data compare in terms of bibliometric standards like WoS or MAG? How are issues like entity resolution and disambiguation resolved?

o Are the publications strictly peer-reviewed research articles? Other datasets like WoS and MAG typically include a litany of other things including proceedings, editorials, cases, reviews, etc.

o Confused about the “two snapshots” description of the data b/c there appears to be a panel dimension in both figures 1 and 2. Thus, I am pretty puzzled about the structure of the underlying data.

• As I see it, there are effectively four possible things changing over the time period: (1) norms in academia, i.e. papers vs. books; (2) the number of people working in various fields; (3) an increasing in “teaming” a la Ben Jones (2021); and (4) individual productivity. I think it would be fruitful to try to hold some of these things constant so we can learn something more definitive. For instance, would the author’s result hold if the author(s) controlled for the fact that people just might be on more papers b/c of “teaming”? What about if the author(s) controlled for productivity? I expect either of these things could easily be done by simply renormalizing the data and would go a long way in terms of convincing me that the proffered explanation is the correct one.

Minor Points:

• Confidence intervals on the figures out be nice.

• Figure 1 is far too busy and the legends are missing

• Bibliometric databases are usually not very good at cataloguing books. Should we be concerned with that?

Reviewer #3: Review of “More journal articles and fewer books: publication practices in the social sciences in the 2010s”

This is an interesting and, as far as I can tell, technically sound analysis.

I think the overall finding is quite striking. My only suggestions are to (1) provide a bit more information on the data generating process (CVs, external databases, or both?) and sampling frame for AA; (2) think about reducing the lit review starting on line 66 a bit (which is a bit long and meandering); (3) up front (and in the abstract) state clearly what the research question(s) is/are and why important. Finally (4) some information on the growth of the number of journals over this period may be useful background, as would any information on changes in the academic book market and/or tenure guidelines.

One may also consider doing some panel analyses to examine “within individual” changes over time. Are the individuals who stopped publishing books now publishing journal articles, or aging out of the publication process (or academe)? This may also help inform some of the compositional discussions on the bottom of page 21 (lines 368-370).

Some discussion of what is gained/lost in the move from books to journals, or a least an overview of the relevant issues, would also be interesting to bring into the last section. Such as discussion (to the extent there is an active debate about the pros/cons) could also help motivate the analyses up front.

6. PLOS authors have the option to publish the peer review history of their article (what does this mean?). If published, this will include your full peer review and any attached files.

Reviewer #1: No

Reviewer #2: No

Reviewer #3: No

---

## [Author Response · Author response to Decision Letter 0]

11 Dec 2021

Dear Dr. Rosenbloom,

We thank you and the reviewers for your thoughtful comments and suggestions, which we believe made this manuscript stronger and more focused. We address the concerns in your letter first, and then turn to each reviewer in order. We list the references added and deleted at the end of this document.

1. All three reviewers ask in their own way for you to clarify what is captured in the Academic Analytics data, addressing how these data are generated and clarifying any potential biases or mis-measurements that result from how the data are generated.

Your comments about further description of the Academic Analytics database are well taken. We added additional description in the subsection “Data Source” at the beginning of the Materials and Methods section, including information on the scope of the database and how it is assembled. We also uploaded a more comprehensive description to the public OSF site for the project (“Expanded Database Description.pdf” at (https://osf.io/2x4uf/). We address the specific issues addressed by the each reviewers in the order they appear, below.

2. Reviewer #2 offers a useful typology of possible sources of change that could be driving the results you document. You should consider whether this framework is helpful, and the extent to which you can distinguish between these causes of change.

We agree with Reviewer #2 that changes in norms or expectations, changes in the number of people working in specific areas, increased co-authorship or teaming, and increases in individual productivity may all contribute to the increase in article publication and decrease in book publication across the social sciences. We performed three additional analyses to explore these potential factors, and detail them in-line in our reply to Reviewer #2, below.

3. Reviewers #1 and #3 suggest ways that you might exploit the panel nature of your data (repeated observations on the same individual at different dates) to help illuminate what is driving the changes you observe.

We agree with both reviewers on this point, and we conducted a new analysis to attempt to address this issue. Specifically, we repeated our analysis using only faculty members who appear in the earliest iteration of the data (2011) and remain in the data throughout the nine-year study timeframe. This new analysis is detailed in the results section.

4. Reviewer #3 suggests being clearer about what your research question is in the abstract and at the start of the article. This reviewer also suggests reducing and tightening the literature review.

We now state our research questions in the abstract, and we also introduce them earlier in the introduction. The literature review was substantially reduced (nearly 900 words shorter), now focusing more explicitly on changes in research environment and research assessment.

 

Reviewer #1

1. The first issue with the paper is that the Academic Analytics commercial database should be described in more detail. As this is a commercial database, it is unlikely to be familiar to most researchers, and without a little bit more detail it is difficult to put the results into context.

We appreciate the reviewer’s request to describe the database in more detail, and have taken two actions to remedy this. We added a new subsection “Data Source” in the Materials and Methods section, describing in more detail the scope of the database and how Academic Analytics assembles the faculty lists and publications data. We also uploaded a more comprehensive description to the public OSF site for the project (“Expanded Database Description.pdf” at https://osf.io/2x4uf/).

2. In particular, should we think of the data as containing the universe of all faculty, only those faculty with a publication, only faculty from a subset of institutions, or only a sample of faculty? Are the same faculty included in multiple waves of the data, and if so, might a panel data structure allow the authors to say more about how an individuals’ publication behaviors are changing over time? The number of departments is rising over time (Table 1). Is this due to the creation of new academic departments, or to the Academic Analytics database obtaining more complete coverage over time? Answering these kinds of questions is important for interpreting the results.

These are excellent suggestions, thank you. In addition to adding text to the “Data Source” subsection defining the scope of universities and faculty members included (Assistant, Associate, and Full professors at all Ph.D. granting institutions in the US), we also performed two additional analyses. We re-ran the analysis including only faculty members that were in the earliest year (2011) and remained in the database for the nine-year study period, to explore whether faculty members individually have changed their publishing behavior or whether faculty members that first appear the database in later years (new hires or newly established departments) account for the changes observed. We now note in the manuscript that we cannot distinguish between departments that are newly established and those that were added to the Academic Analytics database after 2011 but existed before that year, and we re-ran the analysis including only those departments that were in the 2011 data and persisted throughout the study period.

3. Second, the authors make frequent references to new generations of researchers showing different publication patterns. Methodologically, the paper needs some discussion of the age-time-cohort issues: are we observing differences across generations, differences within individuals as they age, or a secular change over time. The fact that both junior and senior faculty publish more papers and fewer books would lead me to believe that this is a secular change in publication practices rather than a generational issue, although the effect is larger for younger researchers. In general, it is not possible to separately identify age versus cohort versus time effects, and so the authors should be even more careful when describing this set of results.

In addition to the analysis described above (comparing only scholars who appear in each iteration of the database), we also clarified the text based on this suggestion. Specifically, the section “Changing publishing practices” now frames the relatively larger increase in articles per person among early career researchers (compared to other age cohorts) within the context of the relatively smaller effect within-individual articles per person from the new analysis.

Reviewer #2

1. An expanded discussion of the dataset is critical. Although the authors cite Academic Analytics as the source, it is unclear whether these data contain both the faculty rosters or publication data or both. Regardless, it seems critical to explain how these data are generated and how the rosters are linked with the bibliometric data. How do these data compare in terms of bibliometric standards like WoS or MAG?

We have added additional information about the database in the body of the text and provided a longer, more detailed description of the database as supplemental materials (https://osf.io/2x4uf/). We also added a citation to Harzing’s (2019) study demonstrating that AcA’s source of journal articles (CrossRef) has similar or greater coverage of publications than Web of Science or Scopus, but is less comprehensive than Google Scholar and Microsoft Academic.

2. Are the publications strictly peer-reviewed research articles? Other datasets like WoS and MAG typically include a litany of other things including proceedings, editorials, cases, reviews, etc

We now note in the “Data source” section that our sample of journal articles is limited to peer-reviewed outputs, excluding, e.g., book reviews, obituaries, and other article types that do not represent original peer-reviewed research.

3. Confused about the “two snapshots” description of the data b/c there appears to be a panel dimension in both figures 1 and 2. Thus, I am pretty puzzled about the structure of the underlying data.

We removed references to “snapshots” (we agree that was confusing and appreciate the suggestion!). We report mainly on comparisons of the earliest (2011) and latest (2019) datasets, but have now made all the intermediate years available publicly on OSF (https://osf.io/2x4uf/) and note in the text that all of the data years were used in preparation of the figures in order to explore temporal trends.

4. As I see it, there are effectively four possible things changing over the time period: (1) norms in academia, i.e. papers vs. books; (2) the number of people working in various fields; (3) an increasing in “teaming” a la Ben Jones (2021); and (4) individual productivity. I think it would be fruitful to try to hold some of these things constant so we can learn something more definitive. For instance, would the author’s result hold if the author(s) controlled for the fact that people just might be on more papers b/c of “teaming”? What about if the author(s) controlled for productivity? I expect either of these things could easily be done by simply renormalizing the data and would go a long way in terms of convincing me that the proffered explanation is the correct one.

This is an excellent summary of factors that may drive the changes we observed. 1) We re-ran our analysis including only faculty members that were in the earliest year (2011) and remained in the database for the nine-year study period, to explore whether faculty members individually have changed their publishing behavior or whether faculty members that first appear the database in later years (new hires or newly established departments) account for the changes observed; 2) we re-ran the analysis including only those departments that were in the 2011 data and persisted throughout the study period; and 3) we calculated the median number of coauthors per article in 2011 and 2019 to explore changes in “teaming” over the study period. Each of these is now discussed in the article and the results presented as supplementary materials. We found that the median number of authors per paper did increase over the study period, and we include a paragraph about this the final section of the paper and a supplemental data table. Our experimental design, however, is not suited to directly control for this phenomenon – we tabulated the total number of unique articles produced by all the scholars within a discipline, regardless of how many scholars within that discipline authored that article. The increasing rates of co-authorship observed by others (and our new analysis) accompanies an increase in the total number of unique articles produced by a discipline’s scholars, but a different experimental design is required to flesh out whether there’s a causal relationship. We suggest this as a future research question.

5. Confidence intervals on the figures out be nice. Figure 1 is far too busy and the legends are missing

We agree, and have removed Figure 1 from the manuscript. The data in the tables in the manuscript and in the supplementary files are sufficient to clearly demonstrate the increase and decrease in article and book publications, respectively.

6. Bibliometric databases are usually not very good at cataloguing books. Should we be concerned with that?

We now include the provider of Academic Analytics’ books data in the “Data Source” section, and we believe the list of publishers covered to be exhaustive for the disciplines examined. We also uploaded the list of all the book publishers included among the supplementary files publicly available at OSF (https://osf.io/2x4uf/).

Reviewer #3

1. provide a bit more information on the data generating process (CVs, external databases, or both?) and sampling frame for AA; 

Thank you for this suggestion, we agree and have made two additions. We include additional descriptions of the data and collection process in the subsection “Data Source” at the beginning of the Materials and Methods section, including information on the scope of the faculty and publications databases and how the data were assembled. We also uploaded a more comprehensive description to the public OSF site for the project (“Expanded Database Description.pdf” at (https://osf.io/2x4uf/).

2. think about reducing the lit review starting on line 66 a bit (which is a bit long and meandering); 

We substantially reduced the literature review and tightened the entire introductory section, it is now more focused and about 900 words shorter.

3. up front (and in the abstract) state clearly what the research question(s) is/are and why important. 

The research questions now appear within the abstract, and have been moved closer to the beginning of the Introduction section.

4. some information on the growth of the number of journals over this period may be useful background, as 

would any information on changes in the academic book market and/or tenure guidelines.

Thank you for this suggestion. We discuss growth in the numbers of journals in the introduction, and we quantify the number of journals represented in our study in 2011 and 2019 early in the results section. We also added a section on the scholarly book market and the possible impact of fewer social science books on libraries.

5. One may also consider doing some panel analyses to examine “within individual” changes over time. Are the individuals who stopped publishing books now publishing journal articles, or aging out of the publication process (or academe)? This may also help inform some of the compositional discussions on the bottom of page 21 (lines 368-370).

This is an excellent addition to the study, the suggestion is much appreciated. We conducted an analysis including only faculty members who were in the earliest year of data (2011) and remained throughout the nine-year study sample. Interestingly, the ratio of books per article was rather stable when this sample was examined, which we believe supports our argument that the overall increase in article production may be attributable to early career researchers (i.e., new hires) producing increasingly greater numbers of articles rather than faculty members shifting from one publication mode to another (or at least ECR’s increase in article authorships is greater than middle-career and senior scholars’ transition from books to articles).

6. Some discussion of what is gained/lost in the move from books to journals, or a least an overview of the relevant issues, would also be interesting to bring into the last section. Such as discussion (to the extent there is an active debate about the pros/cons) could also help motivate the analyses up front

We found little published debate about the pros and cons of publishing books versus journal articles, but new references and discussion have been added as the last two paragraphs of the Discussion section documenting the potential impact on acquisitions librarians and the potential for greater synthesis and contextualization that book publications offer.

We added twelve items to our references list:

1. Johnson R, Watkinson A, Mabe M. The STM Report: an overview of scientific and scholarly publishing - fifth Edition [Internet]. STM: International Association of Scientific, Technical and Medical Publishers; 2018 Oct. Available from: https://www.stm-assoc.org/2018_10_04_STM_Report_2018.pdf

2. Henriksen D. The rise in co-authorship in the social sciences (1980–2013). Scientometrics. 2016 May;107(2):455–76.

3. Jones BF. The Rise of Research Teams: Benefits and Costs in Economics. Journal of Economic Perspectives. 2021 May 1;35(2):191–216.

4. Adams J, Gurney K. Evidence for excellence: has the signal overtaken the substance? : an analysis of journal articles submitted to RAE2008. 2014. 

5. Ostriker JP, Holland PW, Kuh CV, Voytuk JA, editors. A Guide to the Methodology of the National Research Council Assessment of Doctorate Programs [Internet]. Washington, D.C.: National Academies Press; 2009 [cited 2021 Nov 17]. Available from: http://www.nap.edu/catalog/12676

6. Morse R, Castonguay A. How U.S. News Calculated the Best Global Universities Rankings [Internet]. 2021. Available from: https://www.usnews.com/education/best-global-universities/articles/methodology#:~:text=How%20U.S.%20News%20Calculated%20the%20Best%20Global%20Universities,%20%202.5%25%20%209%20more%20rows%20

7. Understanding the Methodology - QS World University Rankings [Internet]. QS Top Universities. 2021 [cited 2021 Oct 25]. Available from: https://www.topuniversities.com/university-rankings-articles/world-university-rankings/understanding-methodology-qs-world-university-rankings

8. Harzing A-W. Two new kids on the block: How do Crossref and Dimensions compare with Google Scholar, Microsoft Academic, Scopus and the Web of Science? Scientometrics. 2019 Jul;120(1):341–9.

9. Cross RL. Digital books and the salvation of academic publishing. Bottom Line. 2011 Nov;24(3):162–6. 

10. Barclay DA. Academic print books are dying. What’s the future_.pdf [Internet]. The Conversation. 2015 [cited 2021 Jul 1]. Available from: https://theconversation.com/academic-print-books-are-dying-whats-the-future-46248

11. Crossik G. Monographs and Open Access: a report to HEFCE [Internet]. HEFCE; 2015 Jan. Available from: https://dera.ioe.ac.uk/21921/1/2014_monographs.pdf

12. Hill SA. Decoupling the academic book. Learned Publishing. 2018 Sep;31:323–7.

We deleted three items from our references list:

1. Altbach PG, de Wit H. Too much academic research is being published [Internet]. University World News. 2018 [cited 2021 Jun 13]. Available from: https://www.universityworldnews.com/post.php?story=20180905095203579

2. Moksony F, Hegedűs R, Császár M. Rankings, research styles, and publication cultures: a study of American sociology departments. Scientometrics. 2014 Dec;101(3):1715–29. 

3. Wolfe A. Books vs. articles: Two ways of publishing sociology. Sociol Forum. 1990 508 Sep;5(3):477–89.

---

## [Decision Letter · Decision Letter 1]

19 Jan 2022

More journal articles and fewer books: Publication practices in the social sciences in the 2010’s

PONE-D-21-27877R1

Dear Dr. Olejniczak,

We’re pleased to inform you that your manuscript has been judged scientifically suitable for publication and will be formally accepted for publication once it meets all outstanding technical requirements.

All three reviewers are clear that you have addressed their concerns about the initial version of the article.  Reviewer #1 does suggest some further potential improvements or clarifications that you should consider as you prepare your manuscript for publication. 

Kind regards,

Joshua L Rosenbloom

Academic Editor

PLOS ONE

Additional Editor Comments (optional):

Reviewers' comments:

Reviewer's Responses to Questions

**Comments to the Author**

1. If the authors have adequately addressed your comments raised in a previous round of review and you feel that this manuscript is now acceptable for publication, you may indicate that here to bypass the “Comments to the Author” section, enter your conflict of interest statement in the “Confidential to Editor” section, and submit your "Accept" recommendation.

Reviewer #1: All comments have been addressed

Reviewer #2: All comments have been addressed

Reviewer #3: All comments have been addressed

2. Is the manuscript technically sound, and do the data support the conclusions?

Reviewer #1: Yes

Reviewer #2: Yes

Reviewer #3: (No Response)

3. Has the statistical analysis been performed appropriately and rigorously? 

Reviewer #1: Yes

Reviewer #2: Yes

Reviewer #3: (No Response)

4. Have the authors made all data underlying the findings in their manuscript fully available?

Reviewer #1: Yes

Reviewer #2: Yes

Reviewer #3: (No Response)

5. Is the manuscript presented in an intelligible fashion and written in standard English?

Reviewer #1: Yes

Reviewer #2: Yes

Reviewer #3: (No Response)

6. Review Comments to the Author

Reviewer #1: Overall, I am quite pleased with the authors revision to the manuscript. The more detailed description of the data is extremely helpful to the reader. I also greatly appreciate the analysis that restricts attention to departments and authors that were in both 2011 and 2019, as this helps to understand what is driving the observed effects.

I would still like to know a bit more about the database. For instance, what fraction of CrossRef articles are successfully matched to academic authors? If the match rate is less than 100%, what is likely causing the failures to match? A brief description of how Baker & Taylor operates and how comprehensive they are would be useful as well. As a reviewer, I always want more information about the data, and while these extra details would be nice to have, to my mind they are not strictly necessary to have a publishable article.

The few comments that are essential to address, in my opinion, are minor issues in the writing to clarifying some of the language to avoid confusion and to accurately reflect the limits of the data and analysis. Specifically:

-Some of the language about whether the effects are driven by within-individual changes or changes in the composition of authors are confusingly worded, especially in the Abstract where there is less context. For instance, the abstract reads, “The increase in article publication and decrease in book publication largely represents individual authors publishing more articles and fewer books, respectively, rather than `book authors’ transitioning to become `article authors’ en masse.” The terms “book authors” and “article authors”, and the idea that one transitions between them, are not intuitive to me. I suspect there may be clearer ways to indicate that the effects are driven by the intensive margin rather than extensive margin. For instance, why not just say something like “Individual authors publish more article and fewer books, but the fraction of authors who publish at least one article increases only slightly and the fraction of authors who publish at least one book decreases only slightly.” (I’m not a stickler for exact wording, but the current terms were confusing to me.)

-The authors note that “Changes in the number of departments in each discipline over the 9-year period may reflect the creation of new departments at the universities studied, but it may also reflect an increased scope of data collection in the AcA database” (pg. 10). But when discussing changes in International Affairs and Public Administration/Policy departments, the authors write that “Growth in both the number of scholars and number of departments in these disciplines suggests that American research universities strategically grew these areas of study both through investment in new faculty hires and by establishing new academic departments” (pg. 15). Given the caveat in the first quote, it doesn’t seem that the authors can conclude anything about what causes changes in the number of scholars and departments and the claim should be qualified accordingly.

Reviewer #2: Excellent work addressing the comments. The revised version of the manuscript is cohesive and makes a nice contribution to the literature.

Reviewer #3: (No Response)

7. PLOS authors have the option to publish the peer review history of their article (what does this mean?). If published, this will include your full peer review and any attached files.

Reviewer #1: No

Reviewer #2: **Yes: **Matthew B. Ross

Reviewer #3: No

---

## [Editor Report · Acceptance letter]

24 Jan 2022

PONE-D-21-27877R1 

More journal articles and fewer books: Publication practices in the social sciences in the 2010’s 

Dear Dr. Olejniczak:

I'm pleased to inform you that your manuscript has been deemed suitable for publication in PLOS ONE. Congratulations! Your manuscript is now with our production department. 

Kind regards, 

on behalf of

Dr. Joshua L Rosenbloom 

Academic Editor

PLOS ONE